# Comparison of Safety and Vector-Specific Immune Responses in Healthy and HIV-Infected Populations Vaccinated with MVA-B

**DOI:** 10.3390/vaccines7040178

**Published:** 2019-11-07

**Authors:** Elvira Couto, Vicenç Diaz-Brito, Beatriz Mothe, Alberto C. Guardo, Irene Fernandez, Ainoa Ugarte, Flor Etcheverry, Carmen E. Gómez, Mariano Esteban, Judit Pich, Joan Albert Arnaiz, Juan Carlos López Bernaldo de Quirós, Christian Brander, Montserrat Plana, Felipe García, Lorna Leal

**Affiliations:** 1Infectious Diseases Department, Hospital Clínic-HIVACAT, IDIBAPS, University of Barcelona, 08036 Barcelona, Spain; jcelvira@gmail.com (E.C.); Alberto.Crespo@bd.com (A.C.G.); IFERNANDEZC@clinic.cat (I.F.); UGARTE@clinic.cat (A.U.); MFETCHEV@clinic.cat (F.E.); LALEAL@clinic.cat (L.L.); 2Infectious Diseases Unit, Parc Sanitari Sant Joan de Deu, 08830 Sant Boi de Llobregat, Spain; vicensdbf@gmail.com; 3Irsicaixa AIDS Research Institute-HIVACAT, Hospital Germans Trias i Pujol, 08916 Badalona, Spain; bmothe@irsicaixa.es (B.M.); cbrander@irsicaixa.es (C.B.); 4Department of Molecular and Cellular Biology, Centro Nacional de Biotecnología, Consejo Superior de Investigaciones Científicas (CNB-CSIC), Campus de Cantoblanco, 28049 Madrid, Spain; cegomez@cnb.csic.es (C.E.G.); mesteban@cnb.csic.es (M.E.); 5Clinical Trial Unit, Hospital Clínic, 08036 Barcelona, Spain; JPICH@clinic.cat (J.P.); JAARNAIZ@clinic.cat (J.A.A.); 6Infectious Diseases Department, Hospital Gregorio Marañón, 28009 Madrid, Spain; juanclopezbq@gmail.com; 7Retrovirology and Viral Immunopathology, AIDS Research Group, IDIBAPS, Hospital Clinic, University of Barcelona, 08036 Barcelona, Spain; MPLANA@clinic.cat

**Keywords:** vaccine, HIV-1, preventive, therapeutic, MVA-B

## Abstract

There are few studies comparing the safety and immunogenicity of the same HIV immunogen in healthy volunteers and HIV-infected individuals. We analyzed demographics, adverse events (AEs), and immunogenicity against vaccinia virus in preventive (RISVAC02, *n* = 24 low-risk HIV-negative volunteers) and therapeutic (RISVAC03, *n* = 20 successfully treated chronically HIV-1-infected individuals) vaccine phase-I clinical trials that were performed with the same design and the same immunogen (modified vaccinia virus Ankara-B: MVA-B). Total AEs were significantly higher in HIV-infected patients (mean AEs/patient 6.6 vs. 12.8 (*p* < 0.01)). Conversely, the number of AEs related to vaccination (AEsRV) was similar between both groups. No grade III or IV AEsRV were observed in either clinical trial. Regarding the immunogenicity, the proportion of anti-vaccinia virus antibody responders was similar in both studies. Conversely, the magnitude of response was significantly higher in HIV-infected patients (median binding antibodies at w8 267 vs. 1600 U/mL (*p* = 0.002) and at w18 666 vs. 3200 U/mL (*p* = 0.003)). There was also a trend towards higher anti-vaccinia virus neutralizing activity in HIV-infected individuals (proportion of responders 37% vs. 63% (*p* = 0.09); median IC50 32 vs. 64 (*p* = 0.054)). This study confirms the safety of MVA-B independent of HIV serostatus. HIV-infected patients showed higher immune responses against vaccinia virus.

## 1. Introduction

Despite the success of HIV prevention strategies, including the use of pre-exposure prophylaxis (PrEP) [1,2], and the continuous improvement in HIV treatment [3], an effective prophylactic vaccine and a cure (or at least a functional cure) are desirable [4]. Initial attempts to develop a vaccine against HIV in the early 1980s were not successful and several clinical trials of vaccines have been conducted, with modest outcomes [5,6]. A better understanding of HIV immunogens and more robust approaches are needed to generate improved candidate vaccines [7,8].

Many HIV-1 vaccine clinical trials have tested the same vectors in HIV-uninfected and infected individuals, but frequently with different schedules or inserts [1,2,3,4]. Overall, the efficacy of recommended vaccines against preventable diseases in HIV-infected patients tends to be lower than in uninfected individuals [5,6,7]. However, there is a dearth of proper studies comparing safety and efficacy in HIV-infected vs. uninfected individuals. Regarding vaccines against HIV-1 infection, Bellino et al. [4] have tested a Tat protein candidate both in HIV-infected and uninfected individuals in two clinical trials conducted with the same lot of vaccine, route of administration, vaccination schedule, and at the same clinical center. They concluded that the safety was similar in these two populations.

The modified vaccinia virus Ankara (MVA) is an attenuated, replication-deficient vaccinia virus that is being tested as a recombinant viral vector for cancer and infectious diseases vaccines with a safe profile and the capacity to induce strong humoral and T-cell responses [8,9]. Adverse events (AEs) associated with MVA are mild and transient, occurring during the first 48 h following vaccination [3]. MVA-B is an HIV-1 vaccine candidate expressing Bx08 monomeric gp120 and the fused IIIB Gag-Pol-Nef (GPN) polyprotein of clade B. Our group performed two clinical trials (RISVAC02 [10] and RISVAC03 [11]) with MVA-B, each with the same schedule, lot, route, and centers in non-HIV-1-infected and HIV-1-infected volunteers, respectively. The conduct of the two randomized, double blind, placebo-controlled phase I studies at the same clinical center represents a unique occasion to compare the safety and vector-specific immune responses in healthy and HIV-infected populations vaccinated with MVA-B.

## 2. Methods

The present substudy analyzes the safety and immunogenicity against vaccinia virus of MVA-B in HIV-uninfected and HIV-infected volunteers in two phase I double-blind placebo-controlled clinical trials: RISVAC02 [10] and RISVAC03 [11], respectively. MVA-B contains the codon-optimized HIV-1 clade B inserts BX08-gp120 and IIIB-GPN. The HIV inserts are placed in the thymidine kinase locus of the viral genome, and during infection gp120 is expressed as a secreted product and GPN as an intracellular fusion polyprotein. MVA-B was generated as described [12] by IDT Biologika GmbH, Germany. In RISVAC02, HIV-uninfected male and female volunteers between 18 and 55 years old, at low risk of HIV-1 infection, with an absence of vaccinia-specific antibodies and no history of previous smallpox vaccination were recruited and randomized to receive MVA-B (*n* = 24) or placebo (*n* = 6). In RISVAC03, HIV-1-infected patients older than 18 years and under successful treatment with a CD4 T cell count >450 cells/mm^3^ were included and randomly allocated (balanced randomization (2:1)) to receive MVA-B (*n* = 20) or placebo (*n* = 10). MVA-B was administered in three intramuscular injections (1 × 10^8^ pfu/dose in 0.5 mL) at weeks 0, 4, and 16. In RISVAC03 an analytical treatment interruption (ATI) was performed in 20 patients (vaccines *n* = 12, placebo n = 6) at week 24 (after the last dose of MVA-B) for 8 weeks. The other 10 participants (vaccines *n* = 8, placebo *n* = 4) started a rollover substudy including disulfiram, then antiretroviral therapy (ART) was discontinued at week 48. In all 30 patients the dynamics of the viral rebound were assessed during the first 12 weeks after ART interruption. ART was resumed when national guideline criteria for the initiation of therapy were reached. For this substudy we only analyzed the results of the patients who had received the vaccine. See Figure 1 for schedule and Figure 2 for participant disposition. Both studies were explained to all patients in detail, and all gave written informed consent. The studies were approved by the institutional ethical review board and by the Spanish Regulatory Authorities.

### 2.1. Safety

In RISVAC02 and RISVAC03 the same specific questionnaire collecting the data of the local and systemic AEs was used for seven days following each immunization. Data on other clinical and laboratory events were collected with an open question at each visit and through routine scheduled investigations, respectively. The investigator stated the relationship to vaccination of each adverse event and its grade of severity based on systems in use at the MRC CTU, and the NIH Division of AIDS.

### 2.2. Immunogenicity

Binding antibodies to Vaccinia Virus (VACV) proteins in serum as well as neutralizing antibodies to VACV were assessed at weeks 0, 8, and 18 in RISVAC02, and at weeks 0, 6, and 18 in RISVAC03 according to standardized operating procedures in the same research laboratory as previously described [10,11] (Figure 1).

### 2.3. Statistical Analysis

Characteristics of the study population and data on immunogenicity were recorded as median (interquartile range (IQT)) or proportions. Comparisons were made using the Mann–Whitney U-test or Chi-square test for quantitative or qualitative variables, respectively. All statistical analyses were performed using the SPSS software version 20 (SPSS Inc., Chicago, IL, USA).

### 2.4. Ethic Issue

All subjects gave their informed consent for inclusion before they participated in the study. RISVAC02 and RISVAC03 studies were conducted in accordance with the Declaration of Helsinki. RISVAC02 protocol was approved by the Ethics Committee of Hospital Clinic de Barcelona (July 12th, 2007) and Hospital Gregorio Marañón de Madrid (April 14th, 2008) (RISVAC02 NCT00679497) and Ministry of Health in Spain (January 28th, 2008). RISVAC03 protocol was approved by the Ethics Committee of Hospital Germans Trias I Pujol de Badalona (March 12th, 2010) on behalf of Ethics Committee of Hospital Clinic de Barcelona and Hospital Gregorio Marañón de Madrid (RISVAC03 NCT01571466) and Ministry of Health in Spain (June 2nd, 2011).

## 3. Results

### 3.1. Clinical Characteristics of Participants

In RISVAC02, 24 participants were allocated to the vaccine arm by randomization. All received the three intramuscular MVA-B injections. Two volunteers were lost to follow-up, one after week 18 and the other after week 20. Twenty-two completed follow-up until week 48. Median (range) of age was 29 (19–48) years old. Of the volunteers, 79% were male (*n* = 19). In RISVAC03, 20 HIV-infected participants were allocated to the vaccine arm by randomization. All received three intramuscular MVA-B injections. There were none lost to follow-up. Median (range) of age was 44 (29–70) years old. Of the volunteers 95% were male (*n* = 19). All of them had an undetectable level of viral load at study entry and the median (IQR) of CD4 T cell count was 701 (581–823) cells/mm^3^. Although no differences were observed in gender between both studies (males RISVAC02: 79% vs. RISVAC03: 95% (*p* = 0.20)), participants in RISVAC03 were significantly older (median 29 vs. 44 years (*p* < 0.01)).

### 3.2. Safety

All participants in the RISVAC02 study were included in the safety analysis. In the 24 vaccinated HIV-uninfected volunteers in RISVAC02, 158 AEs were described, 92% of them (145 events) were grade 1. Only 5 were grade 3 and not related to the vaccination. In RISVAC03 all participants were included in the safety analysis. A total of 257 AEs were reported, 90% (131 events) were grade 1 and 8% grade 2 (21 events). There were 5 severe adverse events (2%) and none of them related to the vaccination. No significant differences in the grade severity of events between groups were observed (*p* = 0.60) (Table 1).

The most common adverse event related to vaccination in both studies was pain at the site of administration; this included 100% of the participants in RISVAC02 and 90% in RISVAC03. This was followed by these systemic adverse events: headache, malaise, and myalgia in 50%, 17%, and 42% of participants, respectively, in RISVAC02, and 35%, 65%, and 54% in RISVAC03. Although no significant differences in the rates of local versus systemic adverse events were found overall (see Table 2), a higher proportion of HIV-1-infected patients showed myalgia (*p* = 0.05) and chills (*p* = 0.02).

Although total AEs were significantly lower in the non–infected patients in RISVAC02 compared with the HIV-infected patients in RISVAC03 (mean (SD) AEs/patient 6.6 (4.3) vs. 12.8 (7.1) (*p* < 0.01)), no differences were observed when AEs related to vaccination (AEsRV) were analyzed (mean (SD) AEs/patient 5.3 (3.7) vs. 6.8 (4.9) (*p* = 0.2), RISVAC02 and RISVAC03, respectively).

### 3.3. Immunogenicity

The proportion of anti-vaccinia virus antibody responders was similar in the RISVAC02 and RISVAC03 studies (responders w8: 92% vs. 100% (*p* = 0.19), respectively, and w18: 100% in both groups). However, the magnitude of response was significantly higher in HIV-infected patients (median (IQR) values w8: 267 (154–866) U/mL vs. 1600 (800–3200) U/mL (*p* = 0.002) and w18: 666 (233–2000) U/mL vs. 3200 (1600–6400) U/mL (*p* = 0.003), RISVAC02 and RISVAC03, respectively) (Figure 3A,B). Those data were confirmed when only the youngest participants of RISVAC03 who did not have previous exposition to vaccinia virus and had undetectable baseline titers were analyzed (Figure 3C,D).

There was also a trend towards higher anti-vaccinia virus neutralizing activity in HIV-infected individuals (proportion of responders 37% vs. 63% (*p* = 0.09); median IC50 32 vs. 64 (*p* = 0.054); RISVAC02 and RISVAC03, respectively).

## 4. Discussion

The present study fills a gap in the information about the safety and immunogenicity of a vaccine in HIV-1-uninfected and infected individuals. Whether AEs in vaccine clinical trials are higher in HIV-infected patients or the same as in the general population is not well established. We found that MVA was safe, consistent with what has been reported in previous studies [3]. No severe AEs related to vaccination were observed in either trial. We believe it is especially relevant that the mean of adverse effects per patient was higher in HIV-infected patients compared with non-HIV-infected patients. It is important to bear in mind that RISVAC03 participants (HIV-infected) were older, and this might explain this difference. Conversely, we did not find any difference in the number of AEs per patient related to the vaccination, although a higher proportion of HIV-1-infected patients suffered myalgia and chills. Bellino et al. [4] performed two clinical trials with Tat protein in HIV-1-uninfected and HIV-1-infected individuals. Finally, although a long-term assessment of AEs has not been performed, it seems that the long-term health condition of both uninfected individuals recruited in RISVAC02 (assessed during a boost with MVA-B four years later) and HIV-1-infected participants in RISVAC03 (controlled in three clinical centers) is good.

Overall, the efficacy of recommended vaccines against preventable diseases in HIV-infected patients tends to be lower than in uninfected individuals [5,6,7]. We decided to compare the efficacy against the viral vector instead of against the insert in these two clinical trials for three reasons. First, it is known that previous contact with the vector could influence the response against the insert [13], although it seems that it was not the case with MVA [8,9]. Second, we could not properly compare the different responses to the HIV insert, considering that RISVAC03 participants were already infected by the virus. Finally, the viral vector analysis would help us to know if there are differences in the specific immune responses induced by a determined vaccine and this could be instrumental in the design of future candidates. Contrary to what we expected, we found that HIV-1-infected patients showed a higher level of response against vaccinia virus than uninfected individuals. It could be argued that previous exposition could explain this difference, but it was maintained even when only the youngest participants of RISVAC03 who did not have previous exposition to vaccinia virus and had undetectable basal titers were analyzed.

Our study had a number of limitations. First, the sample size was small, and some differences could not be detected. Second, even though the same standardized technique was used in the same laboratory, as well as the same equipment, staff, and materials in both studies, we could not totally exclude some bias because the analysis of RISVAC02 was performed four years before RISVAC03. Finally, we could not exclude the potential bias associated with the retrospective studies.

## 5. Conclusions

Our study comparing the safety and immunogenicity against the viral vector of MVA-B in two clinical trials with the same schedule, lot, route, and centers in non-HIV-1-infected and HIV-1-infected volunteers showed that the vaccine was similarly safe in both populations. Surprisingly, the response against vaccinia virus was higher in HIV-infected patients regardless of the basal state. These data could be helpful when interpreting and designing new clinical trials that include poxvirus vectors used as preventive or therapeutic vaccines.

## Figures and Tables

**Figure 1 vaccines-07-00178-f001:**
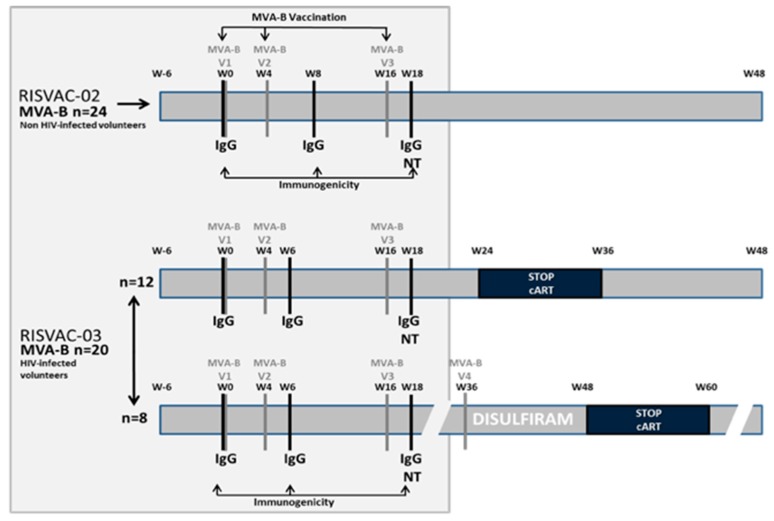
Study design. In this study a comparison of the demographic characteristics, the safety evaluation, and the immunologic response against vaccinia virus (represented inside the grey box) of the 24 non-HIV-infected participants in the modified vaccinia virus Ankara-B: MVA-B arm of the study RISVAC02 against the 20 HIV-infected participants of the MVA-B arm of RISVAC03 was performed. cART: Antiretroviral Therapy. NT: Neutralizing titers.

**Figure 2 vaccines-07-00178-f002:**
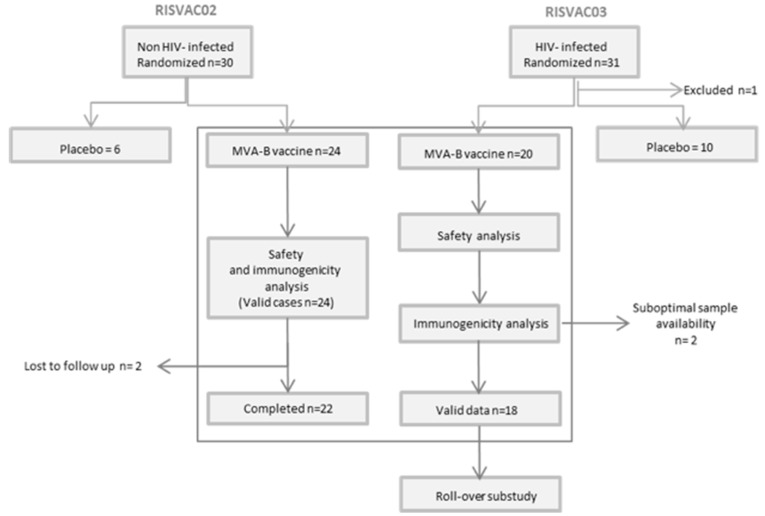
Patient disposition flowchart.

**Figure 3 vaccines-07-00178-f003:**
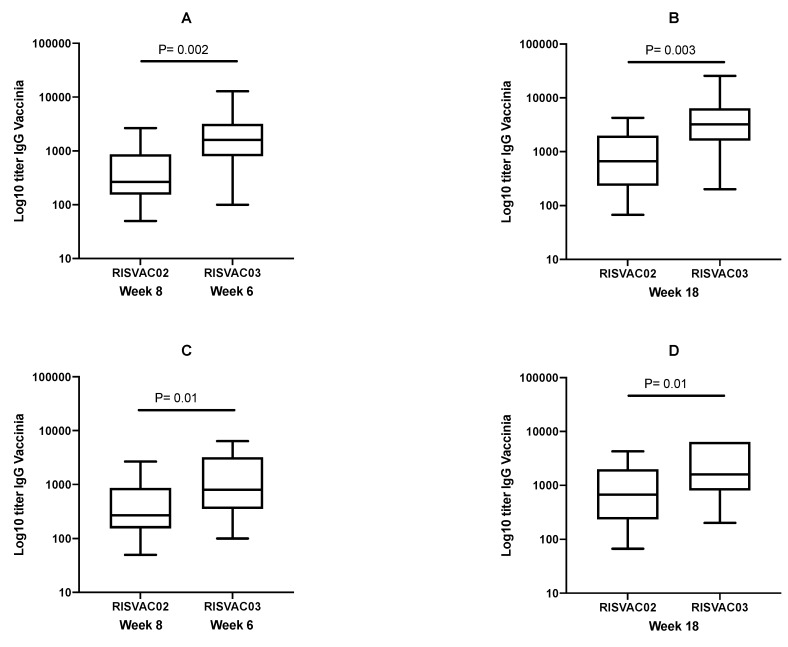
Binding antibodies (IgG titers) against vaccinia virus in RISVAC02 compared to RISVAC03. (**A**) at weeks 8 and 6, respectively; (**B**) at week 18; (**C**) at weeks 8 and 6, respectively, only including participants without previous exposition to vaccinia virus; and (**D**) at week 18, only including participants without previous exposition to vaccinia virus.

**Table 1 vaccines-07-00178-t001:** Description of grade and relation to the vaccine of adverse events (AEs).

**AEs in RISVAC02**	**Related ***	**Non-related ****	**Total**
Grade 1	123	22	145
Grade 2	4	4	8
Grade ≥3	-	5	5
Total	127	31	158
**AEs in RISVAC03**	**Related ***	**Non-related ****	**Total**
Grade 1	102	129	231
Grade 2	13	8	21
Grade ≥3	-	5	5
Total	115	142	257

* Definitely, probably, and possibly related to vaccination. ** Not related or unlikely to be related to vaccination.

**Table 2 vaccines-07-00178-t002:** Number of local and systemic adverse events in the RISVAC02 and RISVAC03 clinical trials.

	RISVAC02	RISVAC03	*p*
LOCAL			
Pain	24 (100)	18 (90)	0.20
Redness	4 (17)	8 (40)	0.12
Itching	4 (17)	6 (30)	0.47
SYSTEMIC			
Headache	10 (42)	7 (35)	0.76
Malaise	12 (50)	13 (65)	0.37
Myalgia	4 (17)	9 (45)	0.05
Nausea/vomiting	3 (13)	2 (10)	1
Chills	1 (4)	7 (35)	0.02
Fever	2 (8)	0 (0)	0.49

N (%).

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
