# Peer review of "Comparison of Safety and Vector-Specific Immune Responses in Healthy and HIV-Infected Populations Vaccinated with MVA-B"

_vaccines, 2019, doi:10.3390/vaccines7040178_

Round 1
Reviewer 1 Report
The article by Elvira Couto et al., analyzed demographics, adverse events (AEs) and immunogenicity against vaccinia virus in a preventive (RISVAC02, n=24 low-risk HIV-1-negative volunteers) and therapeutic (RISVAC03, n=20 successfully treated chronically HIV-1-infected individuals) vaccine phase-I clinical trials that were performed with the same design and the same immunogen (MVA-B) to study the safety and specifically induced immune responses. My comments are listed below:
Given that the preventive (RISVAC02) and therapeutic (RISVAC03) vaccine phase-I clinical trials have been accomplished for several years, do the authors know the current health conditions of the volunteers? Additional discussion would be helpful. In RISVAC02, no volunteer at high risk of HIV-1 infection participated. The authors may wish to comment on this.Author Response
Response: Thank you for the comments of the reviewer. Regretfully, we do not have the current health conditions of all the volunteers who participated in the RISVAC02 (healthy volunteers of low risk of infection, since it was a phase I clinical trial). However, we performed a boost with MVA-B in a subset of these volunteers four years after receiving the first dose of MVA-B and at that time the health condition was optimal (see publication: Guardo AC, et al. Safety and vaccine-induced HIV-1 immune responses in healthy volunteers following a late MVA-B boost 4 years after the last immunization. PLoS One 2017; 12:e0186602). Regarding patients recruited in RISVAC03, We have information that they are in good health because they have been followed-up in the 3 participating hospitals. We have added the following sentence to DISCUSSION: Finally, although a long-term assessment of AEs have not been performed, it seems that the long-term health condition of both uninfected individuals recruited in RISVAC02 (assessed during a boost with MVA-B four years later) and HIV-1 infected participants in RISVAC03 (controlled in 3 clinical centers) is good.
Reviewer 2 Report
GENERAL COMMENTS
1). Lines 2-4: The title should be modified to: Comparison of safety and vector-specific immune responses in healthy and HIV-infected populations vaccinated with MVA-B.
2). The candidate vaccine, MVA-B has previously been evaluated for safety and immunogenicity in clinical studies (Garcia et al., 2011, Vaccine, 29: 8309-8316; Mothe et al., 2015, J Antimicrob Chemother, 70: 1833-1842; Rosas-Umbert et al., 2017; Gomez et al., 2015 e.t.c.). What the current authors have added is the assessment of the immune response to the MVA vector. Therefore, this manuscript should be considered as a short communication and restructured accordingly.
3). Lines 91/92: One of the inclusion criteria for the HIV-infected cohort in RISVAC-03 stated by the authors was a successful treatment with anti-retroviral drugs/a CD T (CD4+ T cells?) cell count > 450 cells/mm3. A CD4+ T-cell count of 500-1600 cells/mm3 is considered normal. In essence, a population of subjects with a CD4+ count > 450 cells/mm3 can be considered “a normal population”. Therefore, the lack of inferiority of the HIV-infected population to the healthy population with regard to vaccinia-specific antibodies in RISVAC-03 would not be unexpected. A comparison of serum samples from the two populations for vaccinia-neutralizing antibodies may help to clarify if qualitative (functional) differences exist in the antibody responses in healthy and HIV-infected subjects. Although the authors indicated on line 121 that neutralization assay was performed, the full data was not shown (lines 182-184). The authors should include data from the neutralization assay.
4). The authors’ Discussion should be more concise.
MINOR COMMENTS
5). Lines 44-47: The two sentences appear contradictory and should be re-written with clarity.
6). Line 69-70: The authors’ statement seems to suggest that MVA was used as a smallpox vaccine during the smallpox eradication campaign. This is not correct. Although MVA was used (primarily in Germany, in the 1970s) as a “primer” in individuals who may be reactive to replication-competent smallpox vaccines, it is not considered one of the mainstream vaccines used in the global eradication of smallpox.
7). Lines 76-79: The authors’ closing introductory statement seems to suggest that the assessment of anti-vector immune responses in the study populations is the primary objective of the study described in the manuscript. This should be made clearer.
8). In lines 77 and 90, the authors statements indicate that the studies described in the manuscript were randomized. However, the authors contradicted this statement in the “Discussion” (line 221-222) by stating “Finally, this was not a randomized clinical trial…..” so which one is it?
9). Lines 54-60: Change “Despite current high efficacy of HIV prophylaxis strategies, as the use of PrEP[1] [2], and the continuous improvement of HIV treatment[3], it is well established that there is still a need for a prophylactic vaccine and for a cure (or at least a functional cure) of the infection[4]. With that aim, since unsuccessful initial approaches in the early 80’s, multiple vaccine trials have been conducted very different strategies [5, 6]. Regretfully, the efficacy of these vaccines have been modest in the scenario and it would be helpful to improve the knowledge about these immunogens to generate potential candidates [7, 8].”
to “Despite the success of HIV prevention strategies, including the use of pre-exposure prophylaxis (PrEP) [1] [2], and the continuous improvement in HIV treatment [3], an effective prophylactic vaccine and a cure (or at least a functional cure) are desirable [4]. Initial attempts to develop a vaccine against HIV in the early 1980s were not successful and several clinical trials of vaccines have been conducted, with modest outcomes [5, 6]. A better understanding of HIV immunogens and more robust approaches are needed to generate improved candidate vaccines [7, 8].
10). Line 67: change “in 2 clinical trials performed with the same schedule, lot, route and center.” to “in two clinical trials conducted with the same lot of vaccine, route of administration, vaccination schedule, and at the same clinical center.”
11). Line 69: change “MVA is a replication-deficient and attenuated vaccinia virus” to “The modified vaccinia virus Ankara (MVA) is an attenuated, replication-deficient vaccinia virus”
12). Line 70: change “of cancer” to “for cancer”
13). Line 71” Replace “incapacity to cause a disseminate infection while induces a strong” with “safety profile and capacity to induce strong”
14). Line 72: “Most of the adverse events (AEs) are mild and transient during the first 48h” to “Adverse events (AEs) associated with MVA are mild and transient, occurring during the first 48 hours”
15). Line 76-77: Replace “The parallel conduction in the same clinical centers” with “The conduct of the two studies at the same clinical center”
16). Line 86: Replace “cell-release” with “secreted”
17). Line 89: Replace “smallpox specific” with “orthopoxvirus-specific” (or “vaccinia-specific”)
18). Lines 122-124: The authors’ statement suggest that neutralization antibodies were evaluated by ELISA. This needs clarification.
19) Line 133: replace “were allocated in to the vaccine arm by randomization” with “were randomized into vaccine arm”
Author Response
1). Lines 2-4: The title should be modified to: Comparison of safety and vector-specific immune responses in healthy and HIV-infected populations vaccinated with MVA-B.
Response: title has been modified as suggested by the reviewer
2). The candidate vaccine, MVA-B has previously been evaluated for safety and immunogenicity in clinical studies (Garcia et al., 2011, Vaccine, 29: 8309-8316; Mothe et al., 2015, J Antimicrob Chemother, 70: 1833-1842; Rosas-Umbert et al., 2017; Gomez et al., 2015 e.t.c.). What the current authors have added is the assessment of the immune response to the MVA vector. Therefore, this manuscript should be considered as a short communication and restructured accordingly.
Response: No problem from our side to consider it as a short communication if the Editors agree
3). Lines 91/92: One of the inclusion criteria for the HIV-infected cohort in RISVAC-03 stated by the authors was a successful treatment with anti-retroviral drugs/a CD T (CD4+ T cells?) cell count > 450 cells/mm3. A CD4+ T-cell count of 500-1600 cells/mm3 is considered normal. In essence, a population of subjects with a CD4+ count > 450 cells/mm3 can be considered “a normal population”. Therefore, the lack of inferiority of the HIV-infected population to the healthy population with regard to vaccinia-specific antibodies in RISVAC-03 would not be unexpected. A comparison of serum samples from the two populations for vaccinia-neutralizing antibodies may help to clarify if qualitative (functional) differences exist in the antibody responses in healthy and HIV-infected subjects. Although the authors indicated on line 121 that neutralization assay was performed, the full data was not shown (lines 182-184). The authors should include data from the neutralization assay.
Response: In fact, these data had already been included in RESULTS as follow: There was also a trend towards higher anti-vaccinia virus neutralizing activity in HIV-infected individuals [proportion of responders 37%vs 63% (p=0.09); median IC50 32 vs 64 (p=0.054); RISVAC02 and RISVAC03, respectively]. Data not shown sentence was a mistake and we deleted it. If the Editors consider that a table or a Figure with the data should be included, we will do it.
4). The authors’ Discussion should be more concise.
Response: As suggested by the Reviewer, the Discussion has been shortened.
MINOR COMMENTS
5). Lines 44-47: The two sentences appear contradictory and should be re-written with clarity.
Response: we have re-written the sentence as follow: Total AEs were significantly higher in HIV-infected patients [mean AEs/patient 6.6 vs 12.8 (p<0.01)]. Conversely, the number of AEs related to vaccination (AEsRV) was similar between both groups.
6). Line 69-70: The authors’ statement seems to suggest that MVA was used as a smallpox vaccine during the smallpox eradication campaign. This is not correct. Although MVA was used (primarily in Germany, in the 1970s) as a “primer” in individuals who may be reactive to replication-competent smallpox vaccines, it is not considered one of the mainstream vaccines used in the global eradication of smallpox.
Response: we have deleted this sentence
7). Lines 76-79: The authors’ closing introductory statement seems to suggest that the assessment of anti-vector immune responses in the study populations is the primary objective of the study described in the manuscript. This should be made clearer.
Response: In the final sentence of the paragraph (and in the title) is expressed that the objective of the study is to compare safety and vector-specific immune responses in healthy and HIV-infected populations vaccinated with MVA-B: The sentence has been changed as follow: The parallel conduction in the same clinical centers of these two randomized, double blinded, placebo-controlled phase I studies represents a unique occasion to compare the safety and vector-specific immune responses in healthy and HIV-infected populations vaccinated with MVA-B.
8). In lines 77 and 90, the authors statements indicate that the studies described in the manuscript were randomized. However, the authors contradicted this statement in the “Discussion” (line 221-222) by stating “Finally, this was not a randomized clinical trial…..” so which one is it?
Response: Both studies were randomized, but the comparisons between both studies (both in safety and immunogenicity) was performed between 2 studies not randomized. To avoid misunderstandings, we have modified the last sentence of the 3rd paragraph of Discussion as follow: Finally, it cannot be excluded the potential bias associated with the retrospective studies.
9). Lines 54-60: Change “Despite current high efficacy of HIV prophylaxis strategies, as the use of PrEP[1] [2], and the continuous improvement of HIV treatment[3], it is well established that there is still a need for a prophylactic vaccine and for a cure (or at least a functional cure) of the infection[4]. With that aim, since unsuccessful initial approaches in the early 80’s, multiple vaccine trials have been conducted very different strategies [5, 6]. Regretfully, the efficacy of these vaccines have been modest in the scenario and it would be helpful to improve the knowledge about these immunogens to generate potential candidates [7, 8].”
to “Despite the success of HIV prevention strategies, including the use of pre-exposure prophylaxis (PrEP) [1] [2], and the continuous improvement in HIV treatment [3], an effective prophylactic vaccine and a cure (or at least a functional cure) are desirable [4]. Initial attempts to develop a vaccine against HIV in the early 1980s were not successful and several clinical trials of vaccines have been conducted, with modest outcomes [5, 6]. A better understanding of HIV immunogens and more robust approaches are needed to generate improved candidate vaccines [7, 8].
Response: The paragraph has been changed as suggested
10). Line 67: change “in 2 clinical trials performed with the same schedule, lot, route and center.” to “in two clinical trials conducted with the same lot of vaccine, route of administration, vaccination schedule, and at the same clinical center.”
Response: The sentence has been changed as suggested
11). Line 69: change “MVA is a replication-deficient and attenuated vaccinia virus” to “The modified vaccinia virus Ankara (MVA) is an attenuated, replication-deficient vaccinia virus”
Response: The sentence has been changed as suggested
12). Line 70: change “of cancer” to “for cancer”
Response: It has been changed as suggested
13). Line 71” Replace “incapacity to cause a disseminate infection while induces a strong” with “safety profile and capacity to induce strong”
Response: The sentence has been changed as suggested
14). Line 72: “Most of the adverse events (AEs) are mild and transient during the first 48h” to “Adverse events (AEs) associated with MVA are mild and transient, occurring during the first 48 hours”
Response: The sentence has been changed as suggested
15). Line 76-77: Replace “The parallel conduction in the same clinical centers” with “The conduct of the two studies at the same clinical center”
Response: The sentence has been changed as suggested
16). Line 86: Replace “cell-release” with “secreted”
Response: It has been changed as suggested
17). Line 89: Replace “smallpox specific” with “orthopoxvirus-specific” (or “vaccinia-specific”)
Response: It has been changed as suggested
18). Lines 122-124: The authors’ statement suggest that neutralization antibodies were evaluated by ELISA. This needs clarification.
Response: It has been deleted and referred to the techniques described in references.
19) Line 133: replace “were allocated in to the vaccine arm by randomization” with “were randomized into vaccine arm”
Response: It has been changed as suggested